**Data Availability Statement:** All relevant data are within the manuscript and its Supporting Information files.

**Funding:** This study did not receive specific funding in any form.

# mRNA Covid-19 vaccines in pregnancy: A systematic review

**Nando Reza Pratama**[1], **Ifan Ali Wafa**[1], **David Setyo Budi**[1], **Manesha Putra**[2], **Manggala Pasca Wardhana**[3], **Citrawati Dyah Kencono Wungu**[4,5]*

**1** Faculty of Medicine, Universitas Airlangga, Surabaya, Indonesia, **2** Division of Maternal-Fetal Medicine, Department of Obstetrics and Gynecology, University of Colorado Anschutz School of Medicine, Aurora, Colorado, United States of America, **3** Department of Obstetrics & Gynecology, Maternal-Fetal Medicine, Dr. Soetomo General Hospital, Faculty of Medicine, Universitas Airlangga, Surabaya, Indonesia, **4** Department of Physiology and Medical Biochemistry, Faculty of Medicine, Universitas Airlangga, Surabaya, Indonesia, **5** Institute of Tropical Disease, Universitas Airlangga, Surabaya, Indonesia

☯ These authors contributed equally to this work.
* citrawati.dyah@fk.unair.ac.id

## Abstract

### Objective

Pregnancy is a known risk factor for severe Coronavirus disease 2019. It is important to develop safe vaccines that elicit strong maternal and fetal antibody responses.

### Methods

Registries (ClinicalTrials.gov, the WHO Clinical Trial Registry, and the European Union Clinical Trial Registry) and databases (MEDLINE, ScienceDirect, Cochrane Library, Proquest, Springer, medRxiv, and bioRxiv) were systematically searched in June 20–22, 2021, for research articles pertaining to Covid-19 and pregnancy. Manual searches of bioRxiv and medRxiv were also conducted. Inclusion criteria were studies that focused on Covid-19 vaccination among pregnant women, while review articles and non-human studies were excluded. Infection rate, maternal antibody response, transplacental antibody transfer, and adverse events were described.

### Results

There were 13 observational studies with a total of 48,039 pregnant women who received mRNA vaccines. Of those, three studies investigated infection rate, six studies investigated maternal antibody response, seven studies investigated antibody transfer, three studies reported local adverse events, and five studies reported systemic adverse events. The available data suggested that the mRNA-based vaccines (Pfizer–BioNTech and Moderna) can prevent future SARS-CoV-2 infection. These vaccines did not show clear harm in pregnancy. The most commonly encountered adverse reactions were pain at the injection site, fatigue, and headache, but these were transient. Antibody responses were rapid after the first vaccine dose. After the booster, antibody responses were stronger and associated with better transplacental antibody transfer. Longer intervals between first vaccination dose and

**Competing interests:** The authors have declared that no competing interests exist.

delivery were also associated with higher antibody fetal IgG and a better antibody transfer ratio.

## Conclusions

The SARS-CoV-2 mRNA vaccines are encouraged for pregnancy. These vaccines can be a safe option for pregnant women and their fetuses. Two vaccine doses are recommended for more robust maternal and fetal antibody responses. Longer latency is associated with higher fetal antibody responses. Further research about its long-term effect on pregnancy is needed.

## Systematic review registration

PROSPERO (CRD42021261684).

## Introduction

Since the beginning of the Covid-19 pandemic, extensive efforts have been made to end this global disaster. One of the most effective approaches is vaccination against severe acute respiratory syndrome Coronavirus 2 (SARS-CoV-2). The efficacy and safety of Coronavirus disease 2019 (Covid-19) vaccines have been demonstrated in adults across a range of demographics [1], but their impact on pregnant women remain unclear due to insufficient information being available. In fact, pregnant women are still at a higher risk of acquiring viral respiratory infections and severe pneumonia due to the unique physiological changes in their immune and cardiopulmonary systems [2, 3]. Although most pregnant women suffer only mild to moderate symptoms, SARS-CoV-2 infection is more severe in pregnant women than in others, with increased risks of hospital admission, intensive care unit stay, and death [4].

Despite their higher risk of SARS-CoV-2 infection, pregnant and lactating women were not included in any initial Covid-19 vaccine trials, resulting in a lack of data to guide vaccine decision-making in these populations [5]. A previous study revealed that most pregnant women with Covid-19 admitted to hospital were asymptomatic, which allows these undetected patients to transmit the virus to others [6–8]. This shows that efforts to prevent SARS-CoV-2 infection, such as by vaccination, are critical for investigations on this population. For this reason, we systematically review the latest evidence on Covid-19 vaccination to summarize its efficacy, immunogenicity, and safety profile in pregnant women.

## Methods

This systematic review adhered to PRISMA (Preferred Reporting Items for Systematic Review and Meta-Analysis) 2020 guidelines [9], and has been registered in the PROSPERO database (CRD42021261684).

### Eligibility criteria

The following study types were included in this review: retrospective, prospective, cohort, randomized controlled trial (RCT), case series, case control, cross-sectional, and crossover. The authors screened the title and abstract of papers independently to identify eligible studies based on the following criteria: (1) pregnant women as subjects; (2) the study involved a Covid-19 vaccine of interest; (3) at least one of our outcomes of interest was reported; and (4)

the paper was written in English. Our primary outcomes included infection rate, maternal antibody titer, and local and systemic adverse events. Our secondary outcomes included neonatal outcome, cord blood antibody titer, and placental transfer ratio. We excluded review articles, irrelevant studies, non-human studies, and duplicates.

## Search strategy and selection of studies

We conducted comprehensive keyword searches, on June 20–22, 2021, to find articles published in trial registries (ClinicalTrials.gov, the WHO Clinical Trial Registry, and the EU Clinical Trial Registry) and databases (MEDLINE, ScienceDirect, Cochrane Library, Proquest, and Springer). We added one eligible study on July 13, 2021. Our research is limited to studies involving humans reported in english. Manual searches, including in bioRxiv and medRxiv, and a bibliographical search were also conducted to obtain additional evidence. The following keywords were used: "[(SARS-CoV-2) OR (Covid-19)] AND [(pregnancy) OR (pregnant)] AND [(vaccine) OR (vaccination)]." Details of the search strategies are available in (**S1 File**). We exported all studies retrieved from the electronic searches into Mendeley reference manager to remove duplicates and perform screening. The two review authors (NRP and IAW) independently screened the titles and abstracts of the articles to identify potentially eligible studies and subsequently screened the full texts independently. Any disagreements between the two review authors were resolved by discussion until a consensus was reached. Excluded studies are described in the PRISMA flow diagram alongside the reasons for their exclusion (**Fig 1**).

## Data extraction

Three authors (NRP, IAW, and DSB) independently extracted relevant data using a structured and standardized form from each selected study. The following information was extracted: first authors' names and publication year, study design, country of origin, sample size, gestational age at first vaccination, sample size, age, sample collection, vaccine type, and outcomes (infection rate, maternal titer antibody, cord blood titer antibody, placental transfer ratio, and local and systemic adverse events). Any disagreements between the review authors were resolved by discussion until a consensus was reached.

## Quality assessment

Two authors (IAW and DSB) independently assessed the risk of bias from each of the included studies using the Newcastle-Ottawa Scale (NOS) assessment tool for cohort studies and the Joanna Briggs Institute (JBI) critical appraisal checklist for case reports, case series, and cross-sectional studies [10, 11]. The NOS contains eight items within three domains, namely, patient selection, comparability, and outcomes. Studies with scores of 7–9, 4–6, and 0–3 were considered to be of high, moderate, and low quality, respectively. Any discrepancies in the scoring were resolved by discussion until a consensus was reached.

## Statistical analysis

Owing to key differences in the comparisons performed in each study and various outcome measures, we could not perform a meta-analysis of the included studies, but instead narratively synthesized the evidence.

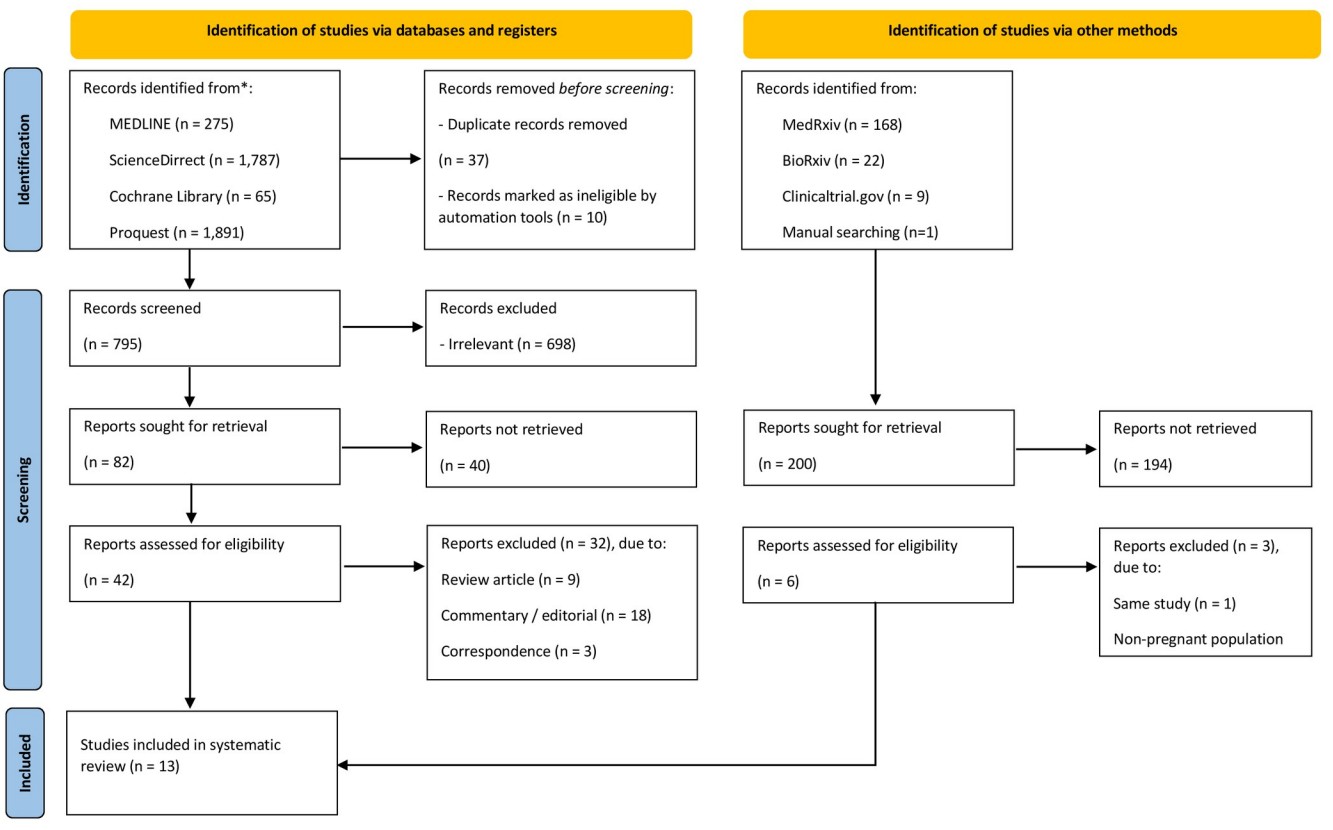

**Fig 1. PRISMA flow diagram of study selection process.**

## Results

### Study selection

The search strategy yielded 4,218 records. After screening the titles and abstracts, 48 potentially eligible articles were reviewed. After full-text assessment, 13 studies were included in this systematic review. The process of study selection in this review is described in the PRISMA flow diagram (**Fig 1**), along with the reasons for exclusion.

### Quality assessment

Nine cohort studies were assessed using the NOS assessment tool and considered to be of high quality (**S1 Table**), while the assessment of the quality of case reports, case series, and cross-sectional studies using the JBI critical appraisal checklist is summarized in **S2–S4 Tables**.

### Study characteristics

A total of 13 observational studies with 48,039 pregnant women who received Covid-19 vaccines were included in this systematic review. Among these studies, 10 (6 cohort, 1 cross-sectional, 1 case series, and 2 case reports) were conducted in the USA, while the other three (all cohort studies) were carried out in Israel in 2021. The detail of study characteristics was presented in **Table 1**.

**Table 1. Characteristics of the included studies.**

| Reference | Study design | Country of origin | Gestational age at first vaccine or 1st vaccine-to-delivery interval dose | Sample size | | Age, y Mean ± SD or Median (IQR) | | Sample collection | Vaccine type | Outcomes |
|---|---|---|---|---|---|---|---|---|---|---|
| | | | | Pregnant (N) | Not Pregnant (N) | Pregnant (N) | Not Pregnant (N) | | | |
| Shimabukuro et al., 2021 [12] | Cohort | United States | NR | Vaccinated 39.870 | 0 | Vaccinated 33 (16–54) | NR | Data from "v-safe and VAERS. | Pfizer–BioNTech and Moderna. | Infection rate, local adverse events, systemic adverse events, pregnancy loss, and neonatal outcomes. |
| Gray et al., 2021 [13] | Cohort | United States | Gestational age at first vaccine (Mean) 23.2 weeks | Vaccinated 84 | Non-pregnant, received vaccine: 16 | Vaccinated 34.1±3.3 | Vaccinated 38.4±8.3 | Antibodies in umbilical cord blood, maternal sera, and breastmilk were quantified using ELISA. Adverse events were assessed using questionnaire. | Pfizer–BioNTech and Moderna. | Maternal antibody titer, local adverse events, systemic adverse events, adverse pregnancy outcome, and composite infant morbidity. |
| Collier et al., 2021 [14] | Cohort | United States | Gestational age at first vaccine dose (N (%)) <14 week: 5 (17%) 14–28 week: 15 (50%) ≥28 week: 10 (33%) | Vaccinated 30. Infected, not vaccinated 22 | Neither pregnant nor lactating, received vaccine: 63 | Vaccinated 35 (32–36) Not vaccinated: 31 (28–36) | Vaccinated: 30 (25–35) Not vaccinated: 34 (33–38) | SARS-CoV-2 RBD in serum and milk were assessed by ELISA. Neutralizing antibody activity was assessed by Luciferase Assay System. | Pfizer–BioNTech and Moderna. | Maternal antibody titer, systemic adverse events, cord blood antibody titer, and breast milk antibody titer. |
| Shanes et al., 2021 [15] | Cohort | United States | 1st vaccine-to-delivery interval (Mean±SD) 45.96±24.3 days | Vaccinated 84. Neither vaccinated nor infected 116 | 0 | Vaccinated 33.7±3.1 Not vaccinated: 32.5±4.8 | NR | Antibody testing from plasma used a paramagnetic particle, chemiluminescent immunoassay. | mRNA vaccines. | Maternal antibody titer and placental finding. |
| Prabhu et al., 2021 [16] | Cohort | United States | NR | Vaccinated 122 | Neonates: 122 | NR | NR | Semi-quantitative testing for RBD used (ET HealthCare) 3 on sera of maternal peripheral blood and neonatal cord blood. | Pfizer–BioNTech and Moderna. | Maternal antibody, Neonatal IgG, Maternal antibody vs neonatal IgG, Placental transfer ratio. |
| Gill and Jones, 2021 [17] | Case Report | United States | Gestational age at first vaccine dose: 32.9 weeks | Vaccinated 1 | 0 | 34 years | NR | Cord blood and maternal blood. | Pfizer–BioNTech | Maternal antibody, cord blood IgG |

*(Continued)*

**Table 1.** (Continued)

| Reference | Study design | Country of origin | Gestational age at first vaccine or 1st vaccine-to-delivery interval dose | Sample size | | Age, y Mean ± SD or Median (IQR) | | Sample collection | Vaccine type | Outcomes |
|---|---|---|---|---|---|---|---|---|---|---|
| | | | | Pregnant (N) | Not Pregnant (N) | Pregnant (N) | Not Pregnant (N) | | | |
| Kadali et al., 2021 [18] | Cross-sectional study | United States | NR | Pregnant, Vaccinated 38 | Non pregnant, received vaccine: 991 | NR | NR | Independent online survey questionnaire (Survey Monkey, San Mateo, CA). Anonymous responses about the side effects were collected from HCWs representing various parts of the country | Pfizer–BioNTech and Moderna. | Local adverse events, and systemic adverse events. |
| Rottenstreich et al., 2021 [19] | Cohort | Israel | 1st vaccine-to-delivery interval (Median (IQR)) 33 (30–37) days | Vaccinated 20 | 0 | 32 (28–37) years | NR | Antibody in maternal and cord blood sera were assessed by chemiluminescent microparticle immunoassay (CMIA). | Pfizer–BioNTech. | Maternal IgG, cord blood IgG and placental transfer ratio. |
| Mithal et al., 2021 [20] | Prospective case series | United States | Gestational age at first vaccine dose (Mean±SD) 33±2 weeks | Vaccinated 27 | Neonates: 28 (1 twin pair) | 33±3 years | NR | Maternal blood and umbilical cord blood using paramagnetic particle, chemiluminescent immunoassay. | Pfizer–BioNTech, Moderna, and unknown | Maternal antibody titer, positive IgM rate, positive IgG rate, IgG transfer outcomes, and Infant IgG outcomes. |
| Theiler et al., 2021 [21] | Cohort | United States | Gestational age at first vaccine dose (Median (IQR)) 32 (13.9–40.6) weeks | Vaccinated 140. Had Covid-19 infection during pregnancy: 212 | 0 | 31.8±3.72 years | 30.0±5.32 years | Electronic medical record from Mayo Clinic | Pfizer–BioNTech and Moderna | Infection rate, Maternal and delivery outcome, and length of stay. |
| Beharier et al., 2021 [22] | Cohort | Israel | Gestational age at first vaccine dose (Mean±SD) 34.5±7.5 weeks | Vaccinated 92 Neither vaccinated nor infected: 66 Past SARS-CoV 2 infections: 74 | 0 | Vaccinated 31.7±5.8 years. Not vaccinated, not infected: 31.6±5.8 years. Past SARS-CoV 2 infections: 28.8±5.8 years | NR | Maternal and fetal blood samples Sera IgG and IgM titers were measured using bead-based multiplex assay (for S1, S2, RBD and nucleocapsid). | Pfizer–BioNTech | Temporal dependence in pregnant people, temporal dependence in neonates, maternal IgG between vaccinated vs PCR-positive, and Maternal-fetal IgG response to infection and vaccination correlation |

(Continued)

**Table 1.** (Continued)

| Reference | Study design | Country of origin | Gestational age at first vaccine or 1st vaccine-to-delivery interval dose | Sample size | | Age, y Mean ± SD or Median (IQR) | | Sample collection | Vaccine type | Outcomes |
|---|---|---|---|---|---|---|---|---|---|---|
| | | | | Pregnant (N) | Not Pregnant (N) | Pregnant (N) | Not Pregnant (N) | | | |
| Paul and Chad, 2021 [23] | Case Report | United States | 36.4 weeks | Vaccinated 1 | 0 | NR | NR | The Electro-chemiluminescence Immunoassay (ECLIA) uses a recombinant protein representing the RBD | Moderna | Cord blood antibody level |
| Goldshtein et al., 2021 [24] | Cohort | Israel | NR | Vaccinated 7,530 Not vaccinated 7,530 | 0 | Vaccinated 31.1±5.01 Not vaccinated: 31.0±4.85 | NR | The Maccabi Healthcare Services database | Pfizer–BioNTech | Infection rate, adverse events, pregnancy outcomes, neonatal outcomes |

Abbreviations: CA (California); ELISA (enzyme-linked immunosorbent assay); HCWs (healthcare workers); Ig (immunoglobulin); IQR (interquartile range); N (number of people); NR (not reported); RBD (receptor binding domain); RBD (receptor binding domain); S1 (spike-1 protein); S2 (spike-2 protein); SD (standard deviation); VAERS (Vaccine Averse Event Reporting System).

## Patient characteristics

The median [interquartile range (IQR)] of the mean or median gestational age across the studies was 32 (31.5–33.2) weeks. All pregnant women reported receiving an mRNA vaccine, either Pfizer–BioNTech or Moderna vaccine, except for four pregnant women who received a vaccine of an unknown type. The median or mean gestational age of the participants at the first vaccination were mostly 30 weeks or more, except in two studies, which reported values of 23.2 weeks and ≤28 weeks. Some studies compared vaccinated pregnant women with unvaccinated pregnant women, either naturally infected or not infected, or vaccinated non-pregnant women.

## Outcomes

Vaccine efficacy was described in terms of the infection rate, which was measured as the proportion of individuals infected with Covid-19. It was reported at different timeframes across the studies. Goldshtein et al. [24] grouped it into ≤10 days, 11–27 days, and ≥ 28 days. Meanwhile, it was grouped into ≤14 days and >14 days by Shimabukuro et al. [12], and Trimesters I, II, and III by Theiler et al. [21]. Immunogenicity was measured in terms of maternal antibody response, fetal antibody response, and transplacental antibody transfer. Safety outcomes were measured as the adverse events, maternal outcomes, and neonatal outcomes. Adverse events were divided into local and systemic, local adverse events included injection-site pain and soreness, while systemic adverse events included fatigue, headache, myalgia, chills, fever, and nausea. Maternal outcomes were divided into pregnancy outcomes and delivery outcomes. The detail of outcomes of the individual studies was presented in **Tables 2 and 3**.

## Infection rate

Three observational studies investigated the infection rate among vaccinated vs. unvaccinated pregnant women [12, 21, 24]. Among pregnant women who received the Pfizer–BioNTech

**Table 2. Outcomes of the individual studies.**

| Reference | Infection rate N (%) | | Maternal SARS-CoV 2 antibody titer (Mean ± SD or Median (IQR)) | | Local adverse event N (%) | |
|---|---|---|---|---|---|---|
| | Intervention | Comparison | Pregnant | Non-Pregnant | Pregnant | Non-Pregnant |
| Shimabukuro et al., 2021 [12] | **Pfizer–BioNTech vaccine** | **Moderna vaccine** | NR | NR | **Pfizer–BioNTech vaccine (1st dose vs 2nd dose)** | NR |
| | ≤14 days after first eligible dose of vaccination: 3 (0.1%). | ≤14 days after first eligible dose of vaccination: 7 (0.4%). | | | Injection-site pain: 7602 (84%) vs 5886 (89%) | |
| | | | | | Injection-site redness: 160 (2%) vs 169 (3%) | |
| | | | | | Injection-side itching: 103 (1%) vs 109 (2%) | |
| | >14 days after first eligible dose of vaccination: 9 (0.4%) | >14 days after first eligible dose of vaccination: 3 (0.2%) | | | **Moderna vaccine (1st vs 2nd dose)** | |
| | | | | | Injection-site pain: 7360 (93%) vs 5388 (96%) | |
| | | | | | Injection-site redness: 348 (4%) vs 491 (9%) | |
| | | | | | Injection-side itching: 157 (2%) vs 193 (3%) | |
| Gray et al., 2021 [13] | NR | NR | NR | NR | **First dose vaccine** | **First dose vaccine.** Injection-site soreness: 12 (75%) Injection site reaction or rash: 0 (0%) **Second dose vaccine.** Injection-site soreness: 12 (75%) Injection site reaction or rash: 0 (0%) |
| | | | | | Injection-site soreness: 73 (88%) Injection site reaction or rash: 1 (1%) | |
| | | | | | **Second dose vaccine** | |
| | | | | | Injection-site soreness: 44 (57%) Injection site reaction or rash: 1 (1%) | |
| Collier et al., 2021 [14] | NR | NR | **Vaccinated** | **Vaccinated** | NR | NR |
| | | | RBD IgG (median): 27,601 AU | RBD IgG (median): 37,839 AU | | |
| | | | Neutralizing Ab (median): 910 AU | Neutralizing Ab (median): 901 AU | | |
| | | | **Infected** | **Infected** | | |
| | | | RBD IgG (median): 1,321 AU | RBD IgG (median): 771 AU | | |
| | | | Neutralizing Ab (median): 148 AU | Neutralizing Ab (median): 193 AU | | |
| Shanes et al., 2021 [15] | NR | NR | **Vaccinated** | NR | NR | NR |
| | | | RBD IgG: 22.8±14.5 | | | |
| | | | RBD IgM: 4.1±13.2 | | | |
| | | | **Unvaccinated** | | | |
| | | | RBD IgG: 0.04±0.05 | | | |
| | | | RBD IgM: 0.19±0.12 | | | |
| Prabhu et al., 2021 [16] | NR | NR | NR | NR | NR | NR |

*(Continued)*

**Table 2.** (Continued)

| Reference | Infection rate N (%) | | Maternal SARS-CoV 2 antibody titer (Mean ± SD or Median (IQR)) | | Local adverse event N (%) | |
|---|---|---|---|---|---|---|
| | Intervention | Comparison | Pregnant | Non-Pregnant | Pregnant | Non-Pregnant |
| Gill and Jones, 2021 [17] | NR | NR | SARS-CoV-2 IgG titer: 1:25600 (+) | NR | NR | NR |
| Kadali et al., 2021 [18] | NR | NR | NR | NR | Sore arm or pain: 37 (97%) Itching: 2 (5%) Muscle spasm: 1 (3%) | Sore arm or pain: 894 (90%). Itching: 98 (10%). Muscle spasm: 103 (10%) |
| Rottenstreich et al., 2021 [19] | NR | NR | Anti-S IgG: 319 (211–1033) AU/mL Anti-RBD-Specific IgG: 11,150 (6154–17,575) AU/mL | NR | NR | NR |
| Mithal et al., 2021 [20] | NR | NR | NR | NR | NR | NR |
| Theiler et al., 2021 [21] | **Vaccinated:** None: 138 (99%) Trimester 1: 0 (0%) Trimester 2: 2 (1%) Trimester 3: 0 (0%) **Vaccinated vs Non-vaccinated:** 2 (1.4%) vs 210 (11.3%), p = 0.0004 | **Not vaccinated:** None: 1652 (89%) Trimester 1: 26 (1%) Trimester 2: 84 (5%) Trimester 3: 100 (5%) | NR | NR | NR | NR |
| Beharier et al., 2021 [22] | NR | NR | NR | NR | NR | NR |
| Paul and Chad, 2021 [23] | NR | NR | NR | NR | NR | NR |
| Goldshtein et al., 2021 [24] | **Vaccinated** Cumulative infection: 108 **Vaccinated vs Non-vaccinated:** ≤10 days: 70 (0.93%) vs 73 (0.97%) p = 0.89 11–27 days: 38 (0.51%) vs 83 (1.12%) p<0.01 ≥28days: 10 (0.21%) vs 46 (0.96%) p<0.01 | **Non-vaccinated** Cumulative infection: 202 | NR | NR | NR | NR |

Abbreviations: AU (arbitrary unit); Ig (immunoglobulin); IQR (interquartile range); mL (milliliter); N (number of people); NR (not reported); RBD: (receptor binding domain); SD (standard deviation).

vaccine, 0.1% (3/2,136) suffered Covid-19 infection within 14 days from the vaccination and 0.4% (9/2,136) did so more than 14 days after the vaccination. Additionally, among pregnant women who received the Moderna vaccine, as many as 0.4% (7/1,822) and 2% (3/1,822) suffered Covid-19 infection within 14 days and more than 14 days from the vaccination, respectively [12]. Vaccination significantly reduced the risk of future infection (p = 0.0004) and all cases of infection reported in the first trimester among vaccinated people occurred prior to the

**Table 3. Outcomes of the individual studies.**

| Reference | Systemic adverse events N (%) | | Others |
|---|---|---|---|
| | **Pregnant** | **Non-Pregnant** | |
| Shimabukuro et al., 2021 [12] | **Pfizer–BioNTech vaccine (1st dose vs 2nd dose)** Fatigue: 2406 (27%) vs 4231 (64%). Headache: 1497 (17%) vs 3138 (47%). Myalgia: 795 (9%) vs 2916 (44%). Chills: 254 (3%) vs 1747 (26%). Fever or felt feverish: 256 (3%) vs 1648 (25%). Measured temperature $\geq$ 38˚C: 30 (0%) vs 315 (5%). Nausea: 492 (5%) vs 1356 (20%).

**Moderna vaccine (1st vs 2nd dose).** Fatigue: 2616 (33%) vs 4541 (81%). Headache: 1581 (20%) vs 3662 (65%). Myalgia: 1167 (15%) vs 3722 (66%). Chills: 442 (6%) vs 2755 (49%). Fever or felt feverish: 453 (6%) vs 2594 (46%). Measured temperature $\geq$ 38˚C: 62 (1%) vs 664 (12%). Nausea: 638 (8%) vs 1909 (34%) | NR | **Maternal and delivery outcomes.** Pregnancy loss among complete pregnancy, N (%):<br>• Abortion: 104 (12.6%)<br>• Stillbirth: 1 (0.1%)<br>Neonatal outcome among live-born infants, N (%):<br>• Preterm birth: 60 (9.4%)<br>• Small size for gestational age: 23 (3.2%)<br>• Congenital anomalies (N = 16; 2.2%)<br>• Neonatal death (N = 0; 0%) |
| Gray et al., 2021 [13] | **First dose vaccine.** Headache: 7 (8%). Muscle aches: 2 (2%). Fatigue: 12 (14%). Fever or chills: 1 (1%).

**Second dose vaccine.** Headache: 25 (32%). Muscle aches: 37 (48%). Fatigue: 41 (53%). Fever or chills: 25 (32%) | **First dose vaccine (N (%)).** Headache: 5 (31%). Muscle aches: 2 (12%). Fatigue: 6 (38%). Fever or chills: 1 (6%).

**Second dose vaccine (N (%)).** Headache: 6 (38%). Muscle aches: 7 (44%). Fatigue: 9 (56%). Fever or chills: 8 (50%) | **Adverse pregnancy outcome, N (%).** Fetal growth restriction: 0 (0%). Preeclampsia/gestational hypertension: 0 (0%). Preterm delivery (spontaneous): 1 (8%). Preterm delivery (medically indicated): 0 (0%) **Composite infant morbidity, N (%).** Supplemental oxygen/CPAP: 1 (8%). TTN: 1 (8%). Special care nursery admission: 0 (0%). NICU admission: 2 (15%). Respiratory distress syndrome: 0 (0%). Necrotizing enterocolitis: 0 (0%). Sepsis: 0 (0%). Assisted ventilation: 0 (0%). Seizure: 0 (0%). Grade 3/4 intraventricular hemorrhage: 0 (0%). Death: 0 (0%).<br>**IgG Spike response.** Pregnant V1 vs Pregnant V0: $p<0.0001$. Pregnant V2 vs Pregnant V0: $p<0.0001$. Pregnant V2 vs Pregnant V1: $p<0.05$. Cord blood IgG titer vs time from maternal V2 corr.(r): 0.8; $p = 0.01$. Vaccinated pregnant vs natural infection pregnant titer: $p<0.0001$. **IgG RBD response.** Pregnant V1 vs Pregnant V0: $p<0.01$. Pregnant V2 vs Pregnant V0: $p<0.0001$. Pregnant V2 vs Pregnant V1: $p<0.001$. Cord blood IgG titer vs time from maternal V2 corr.(r): 0.50; $p = 0.17$. **Neutralizing antibody titer (umbilical cord vs maternal serum).** Medial (IQR): 104 (61.2–188.2) vs 52.3 (11.7–69.6); $p = 0.05$<br>**Antibodies transfer from maternal to cord blood.** Spike IgG3 (r): 0.93; $p = 0.03$. RBD IgG3 (r): 0.81; $p = 0.07$ |
| Collier et al., 2021 [14] | Fever after first dose: 0 (0%). Fever after second dose: 4 (14%) | Fever after first dose: 1 (2%). Fever after second dose: 27 (52%) | **RBD IgG titer median (mother serum vs cord blood).** Vaccinated: 14,953 vs 19,873. Infected: 1,324 vs 635.<br>**Neutralizing antibodies titer median (mother serum vs cord blood)** Vaccinated: 1,016 AU vs 324. Infected: 151 vs 164.<br>**RBD IgG against SARS-CoV-2 Variants of Concern.** Binding antibody responses were comparable against wild type USA-WA1/2020 and B.1.1.7 RBD proteins in nonpregnant, pregnant, and lactating women and in infant cord samples but were lower for the B.1.351 RBD protein. |
| Shanes et al., 2021 [15] | NR | NR | NR |
| Prabhu et al., 2021 [16] | NR | NR | **Positive maternal antibody rate**<br>Women with detectable:<br>• IgG & IgM (N (%)): 87 (71%)<br>• IgG only (N (%)): 19 (16%)<br>Women with no detectable IgG & IgM (N (%)): 16 (13%)<br>**Positive neonatal antibody rate**<br>IgG from whom the mother received:<br>• One vaccine dose (N (%)): 24 (43.6%)<br>• Two vaccine doses (N (%)): 65 (98.5%)<br>**Placental transfer outcome**<br>Maternal IgG and neonatal IgG correlation (R): 0.89, $p<2.2$ e-16<br>Placental transfer ratio and weeks elapsed since maternal vaccination dose 2 correlation (R): 0.8, $p = 2.6$ e-16 |

(*Continued*)

**Table 3.** (Continued)

| Reference | Systemic adverse events N (%) | | Others |
|---|---|---|---|
| | **Pregnant** | **Non-Pregnant** | |
| Gill and Jones, 2021 [17] | NR | NR | **Cord blood antibody**: SARS-CoV-2 IgG titer: 1:25600 (+) |
| Kadali et al., 2021 [18] | **Pregnant.** Fatigue: 22 (58%). Headache: 19 (50%). Chills: 18 (47%). Myalgia: 13 (34%). Nausea: 11 (29%). Fever: 6 (16%). Seizure*: 1 (3%) | **Non pregnant.** Fatigue: 643 (65%). Headache: 519 (52%). Chills: 424 (43%). Myalgia: 488 (49%). Nausea: 211 (21%). Fever: 279 (28%). Seizure*: 0 (0%) | NR |
| Rottenstreich et al., 2021 [19] | NR | NR | **Cord-Blood level.** Anti-S IgG, median (IQR): 193 (111–260) AU/mL. Anti-RBD-specific IgG, median (IQR): 3494 (1817–6163) AU/mL. <br> **Placental transfer ratio.** Anti-S IgG, median (IQR): 0.44 (0.25–0.61). Anti-RBD-specific IgG, median (IQR): 0.34 (0.27–0.56) |
| Mithal et al., 2021 [20] | NR | NR | **Positive IgM rate.** Maternal serum, N (%): 15 (56%). Cord blood, N (%): 0 (0%). <br> **Positive IgG rate** <br> Maternal serum, N (%): 26 (96%). Cord blood, N (%): 25 (89%) <br> **IgG transfer outcomes** <br> Maternal to infant, mean±SD: 1.0±0.6. Latency from vaccination to delivery vs IgG transfer ratio correlation, β: 0.2 (95%CI 0.1–0.2) <br> **Infant IgG outcomes** <br> Having received the 2$^{nd}$ vaccine dose vs infant IgG level correlation, β: 19.0 (95%CI 7.1–30.8). Latency from vaccination to delivery vs infant IgG level correlation, β: 2.9 (95%CI 0.7–5.1) |
| Theiler et al., 2021 [21] | NR | NR | **Maternal and delivery outcome, N (%) (vaccinated vs unvaccinated)** <br> AOI: 91 (5%) vs 7 (5%); $p = 0.9524$. AOI excluding laceration: 55 (3%) vs 5 (4%); $p = 0.6071$. Hypoxic, ischaemic encephalopathy: 1 (0%) vs 0 (0%); $p = 1$. Uterine rupture, AOI: 1 (0%) vs 0 (0%); $p = 1$. Unplanned ICU admission: 2 (0%) vs 1 (1%); $p = 0.1956$. Birth trauma: 11 (1%) vs 0 (0%); $p = 1$. Return to OR: 6 (0%) vs 1 (1%); $p = 0.3985$. NICU admit > 2500g: 11 (1%) vs 1 (1%); $p = 0.5821$. 5 Minute Apgar <7: 38 (2%) vs 3 (2%); $p = 0.7617$. Hemorrhage with transfusion: 5 (0%) vs 1 (1%); $p = 0.3531$. Third- or fourth-degree laceration: 37 (2%) vs 2 (1%); $p = 1$ <br> Mode of delivery: $p = 0.6517$ <br> • spontaneous vaginal: 1238 (66%) vs 89 (64%) <br> • operative vaginal: 69 (4%) vs 7 (5%) <br> • cesarean: 555 (30%) vs 44 (31%) <br> Gestational age delivery: $p = 0.7028$ <br> • 37+: 1703 (91%) vs 127 (91%) <br> • 32–36.9: 134 (7%) vs 10 (7%) <br> • 24–31.9: 21 (1%) vs 2 (1%) <br> • <24: 4 (0%) vs 1 (1%) <br> Quantitative blood loss > 1000mL: 56 (3%) vs 6 (4%); $p = 0.4452$. Transfusion: 241 (13%) vs 25 (18%); $p = 0.1198$. Thromboembolism: 2 (0%) vs 0 (0%); $p = 1$. Stroke: 1 (0%) vs 0 (0%); $p = 1$. Eclampsia/pre-eclampsia (+/-72 h. of delivery): 23 (1%) vs 1(1%); $p = 1$. Gestational hypertension: 225 (12%) vs 19 (14%); $p = 0.6038$. Low birth weight (<2,500g): 121 (6%) vs 11 (8%); $p = 0.5321$. Very low birth weight (<1500g): 21 (1%) vs 3 (2%); $p = 0.2332$. Stillbirth: 6 (0%) vs 0 (0%); $p = 1$ |

(*Continued*)

**Table 3.** (Continued)

| Reference | Systemic adverse events N (%) | | Others |
|---|---|---|---|
| | **Pregnant** | **Non-Pregnant** | |
| Beharier et al., 2021 [22] | NR | NR | **Temporal dependence in pregnant people.** After infection: A gradual rise in IgG humoral response (Anti- S1, S2, RBD and Nucleocapsid) was detected during the first 45 days after infection. After the first dose: In the same period, vaccinated participants receiving the first BNT162b2 dose showed a rapid IgG response to S1, S2, RBD but not Nucleocapsid, resulting in high titer values by day 15 after the first dose. After the second dose: A further rise in IgG was observed following the second dose. **Temporal dependence in neonates** After the first dose: The temporal dependence of fetal IgG for S1, S2 and RBD after vaccination trailed after the maternal IgG showing a significant response already by day 15. After the second dose: A further increase was observed following the second vaccination dose. **Maternal IgG between vaccinated vs PCR-positive.** S1 IgG: higher in vaccination ($p = 0.0009$). RBD IgG: higher in vaccination ($p = 0.0045$). S2 IgG: higher in PCR-positive ($p = 0.0016$). Nucleocapsid IgG: higher in PCR-positive ($p<0.0001$). **Maternal to fetal IgG transfer ratio for S1, S2, RBD, and N.** PCR-positive vs Nucleocapsid-negative group: Significant differences were found for S1, S2 and RBD ($p<0.0002$). PCR-positive vs Nucleocapsid-positive group: For all antibodies did not differ ($p = 0.4577$) **Maternal-fetal IgG response to infection and vaccination correlation.** S1 IgG: $R^2 = 0.9443$; Adjusted $R^2 = 0.9438$; $p<0.0001$. S2 IgG: $R^2 = 0.9353$; Adjusted $R^2 = 0.9348$; $p<0.0001$. RBD IgG: $R^2 = 0.9200$; Adjusted $R^2 = 0.9194$; $p <0.0001$. Nucleocapsid IgG: $R^2 = 0.9366$; Adjusted $R^2 = 0.9361$; $p<0.0001$. Infection vs vaccination maternal-fetal IgG response S1 IgG: $p = 0.2936$ S2 IgG: $p = 0.4212$ RBD IgG: $p = 0.09702$ Nucleocapsid IgG: $p = 0.7616$ |
| Paul and Chad, 2021 [23] | NR | NR | **Cord blood Antibody.** IgG concentration: 1.31 U/mL |
| Goldshtein et al., 2021 [24] | Headache (n = 10, 0.1%). General weakness (n = 8, 0.1%). Stomachache (n = 5, <0.1%). Nonspecified pain (n = 6, <0.1%). Dizziness (n = 4, <0.1%). Rash (n = 4, <0.1%) | NR | **Pregnancy outcomes, N (%) (vaccinated vs unvaccinated).** Abortion: 128 (1.7%) vs 118 (1.6%). Preeclampsia: 20 (0.3%) vs 21 (0.3%). Obstetric pulmonary embolism: 0 vs 0. Maternal death: 0 vs 0 **Neonatal outcomes, N (%) (vaccinated vs unvaccinated).** Intra Uterine Growth Restriction: 36 (0.5%) vs 38 (0.5%). Stillbirth: 1 (<0.1) vs 2 (<0.1). Birthweek: 39 (38–40) vs 39 (38–40). Preterm birth (<37week): 77/1387 (5.6%) vs 85/1427 (6.0%). Infant weight (kg), median (IQR): 3.2 (2.9–3.6) vs 3.2 (2.9–3.5) |

Abbreviations: AU (arbitrary unit); AOI (adverse outcomes index); CPAP (continuous positive airway pressure); NICU (neonatal intensive care unit); ICU (intensive care unit); Ig (immunoglobulin); kg (kilograms); N (number of people); NR (not reported); TTN (transient tachypnea of the newborn); RBD (receptor binding domain); S1 (spike-1 protein); S2 (spike-2 protein); V0 (at the time of first vaccine dose/baseline); V1 (at the time of the second dose/prime profile); V2 (2–6 weeks after the second vaccine dose/boost profile).

* It reached statistical significance (p = 0.0369). However, the participant with a report of seizure has a known history of seizure disorder and her anticonvulsant blood level was reported as borderline low.

first vaccination [21]. There was no significant reduction of the risk of infection within 10 days from vaccination ($p = 0.79$), but risk reduction reached statistical significance 11–27 days after vaccination ($p < 0.001$), and 28 days or more after vaccination ($p < 0.001$) [24].

## Maternal antibody response

Maternal antibody responses were investigated by six observational studies [13–16, 19, 22]. Vaccination induced immunoglobulin (Ig)G and IgM production in 71% (87/122) of pregnant women, 16% (19/122) of pregnant women produced only IgG, while in 13% (16/122) neither IgG nor IgM was detectable [16]. Vaccination provided a rapid immunological response after the first dose, while infection provided a gradual immunological response. Moreover, the administration of a second dose further increased the IgG level among vaccinated women [13, 22]. Spike and receptor-binding domain (RBD) IgG titers rose rapidly after the first dose ($p < 0.0001$ and $p < 0.01$, respectively); in addition, after receiving the booster, they became higher than after the first dose ($p < 0.05$ and $p < 0.001$, respectively) [13].

Vaccination elicited IgG responses against spike (S)1, S2, and RBD, but not nucleocapsid (N) protein. Meanwhile, infection elicited all IgG responses against S1, S2, RBD, and N protein. Although S1 IgG, S2 IgG and RBD IgG responses were observed in both vaccinated and infected pregnant women, the S1 IgG and RBD IgG levels in vaccinated pregnant women were higher ($p = 0.0009$ and $p = 0.0045$, respectively), while S2 IgG and N IgG were higher in infected pregnant women ($p = 0.0016$ and $p < 0.0001$, respectively) [22]. Meanwhile, Gray et al. (2021) reported that spike IgG titer was higher upon vaccination than upon natural infection in pregnant women [13].

Maternal SARS-CoV-2 spike protein IgG levels were 22.8 ± 14.5 AU and 0.04 ± 0.05 AU in vaccinated and uninfected unvaccinated pregnant women, respectively ($p < 0.001$). Meanwhile, the IgM levels were 4.1 ± 13.2 AU and 0.19 ± 0.12 AU, respectively ($p = 0.001$) [15]. Among pregnant women who received two vaccine doses, the median concentration of anti-spike-protein IgG was 319 (211–1,033) AU/mL and the median anti-RBD-specific IgG concentration was 11,150 (6,154–17,575) AU/mL [19]. Meanwhile, in two women who only received one dose of vaccine, the anti-spike-protein IgG concentrations were 50 and 52 AU/mL, while the anti-RBD-specific IgG concentrations were 293 and 1,137 AU/mL [19].

Antibody responses in pregnant and non-pregnant women were evaluated by Collier et al. (2021). They reported the median IgG levels in vaccinated and infected pregnant women. The RBD IgG titers were 27,601 and 1,321, while the neutralizing antibody titers were 910 and 148, respectively. The median RBD IgG titers in vaccinated and infected non-pregnant women were 37,839 and 771, while for the neutralizing antibody they were 901 and 193, respectively [14].

## Transplacental antibody transfer

Seven observational studies investigated transplacental antibody transfer [14, 16, 17, 19, 20, 22, 23]. A prospective case series reported that IgG was detected in 89% (25/28) of cord blood, but no cases had detectable IgM [20]. Moreover, antibody against SARS-CoV-2 RBD and neutralizing antibody were observed in cord blood. In vaccinated women, the maternal and cord blood RBD IgG levels were 14,953 AU and 19,873 AU, while for the neutralizing antibody they were 1,016 AU and 324 AU, respectively [14]. IgG against S protein was also detected in cord blood, with a concentration of 193 (111–260) AU/mL, and its transfer ratio was 0.44 (0.25–0.61). Furthermore, the concentration of IgG against RBD was 3,494 (1,817–6,163) AU/mL and its transfer ratio was 0.34 (0.27–0.56) [19].

Two different case reports described a mother who received two doses of BNT162b2 vaccine and a mother who received one dose of Moderna vaccine; they reported that SARS-CoV-2-specific IgG was detected in maternal blood and cord blood at a titer 1:25,600, while the cord blood IgG concentration was determined to be 1.31 U/mL [17, 23].

Regarding the number of doses received, antibody was detected in 98.5% (65/67) of neonates whose mothers had received two doses of vaccine. However, antibody was detected in

only 43.6% (24/55) of neonates whose mothers had received only one vaccine dose [16]. Receiving the second vaccine dose was positively correlated with infant IgG level [β = 19.0 (95% CI 7.1–30.8)] [20]. In addition to the doses, the interval from vaccination to delivery was correlated with the IgG transfer ratio and infant IgG level. Increased latency from vaccination to delivery was positively correlated with IgG transfer ratio [β = 0.2 (95% CI 0.1–0.2)] and infant IgG level [β = 2.9 (95% CI 0.7–5.1)] [20]. For maternal-fetal IgG response, there was no statistically significant difference between vaccination and SARS-CoV-2 infection for S1 IgG (*p* = 0.2936), S2 IgG (*p* = 0.4212), RBD IgG (*p* = 0.09702), and N IgG (*p* = 0.7616) [22].

## Local adverse events

Three observational studies reported local adverse events [12, 13, 18]. Among pregnant women, injection-site pain was the most common adverse event for both the Pfizer–BioNTech and Moderna vaccines. Following the Pfizer–BioNTech vaccination, as many as 84% (7,602/ 9,052) in the first dose and 89% (5,886/6,638) in the second dose experienced injection-site pain. Meanwhile, for the Moderna vaccine, 93% (7,360/7,930) and 96% (5,388/5,635) experienced injection-site pain following the first and second doses, respectively [12]. It was also reported that 88% (73/84) of pregnant women experienced injection-site soreness following the first vaccination and 57% (44/84) did so following the second dose. Additionally, 75% (12/ 16) of non-pregnant women experienced injection-site soreness after the first and second doses of the vaccine [13].

In pregnant and non-pregnant women, sore arms or pain were observed in 97% (37/38) of pregnant women and 90% (894/991) of non-pregnant women following the Pfizer–BioNTech and Moderna vaccinations [18].

## Systemic adverse events

Systemic adverse events were reported in five observational studies [12–14, 18, 24]. Following first and second vaccinations with the Pfizer–BioNTech vaccine, the six most common systemic adverse events were fatigue [27% (2,406/9,052) vs. 64% (4,231/6,638)]; headache [17% (1,497/9,052) vs. 47% (3,138/6,638)], myalgia [9% (795/9,052) vs. 44% (2,916/6,638)], chills [3% (254/9,052) vs. 26% (1,747/6,638)], fever [3% (256/9,052) vs. 25% (1,648/6,638)], and nausea [5% (492/9,052) vs. 20% (1,356/6,638)]. For the Moderna vaccine, they were fatigue [33% (2,616/7,930) vs. 81% (4,541/5,635)], headache [20% (1,581/7,930) vs. 65% (3,662/5,635)], myalgia [15% (1,167/7,930) vs. 66% (3,722/5,635)], chills [6% (442/7,930) vs. 49% (2,755/ 5,635)], fever [6% (453/7,930) vs. 46% (2,594/5,635)], and nausea [8% (638/7,930) vs. 34% (1,909/5,635)]. Numerically, the incidence of each event was higher for the second dose. Moreover, the Moderna vaccine had more systemic adverse events than the Pfizer–BioNTech vaccine [12]. Seizure was reported in a woman who received the mRNA vaccine (p = 0.0369), but this patient had a history of seizure disorder and the anticonvulsant level in the blood was borderline low [18].

## Maternal outcomes

Maternal outcomes were described as pregnancy outcomes and delivery outcomes. Compared with the findings in unvaccinated pregnant women, vaccination did not significantly affect pregnancy or delivery outcomes. Between the groups, there were no statistically significant difference in pregnancy outcomes such as eclampsia/pre-eclampsia (*p* = 1), gestational hypertension (*p* = 0.6038), gestational age (*p* = 0.7028), and incidence of thromboembolism (*p* = 1) [21]. Moreover, the abortion rate and preterm birth were reported to be 12.6% (104/827) and 9.4% (60/636), respectively [12]. Statistically, vaccination also did not affect delivery outcomes

such as birth trauma ($p = 1$), uterine rupture ($p = 1$), unplanned ICU admission ($p = 0.1956$), blood loss >1,000 mL ($p = 0.4452$), hemorrhage with transfusion ($p = 0.3531$), and mode of delivery ($p = 0.6517$) [21]. Other outcomes were also reported among vaccinated and unvaccinated pregnant women, namely, abortion [1.7% (128/7,530) vs. 1.6% (118/7,530)] and pre-eclampsia [0.3% (20/7,530) vs. 0.3% (21/7,530)]. No cases of obstetric pulmonary embolism or maternal death occurred in either group [24].

## Neonatal outcomes

The effects of mRNA vaccines on neonatal outcomes were reported by four observational studies [12, 13, 21, 24]. As many as 15% (2/13) of cases required NICU admission, 8% (1/13) experienced TTN, and 8% (1/13) required supplemental oxygen or CPAP. Preterm delivery was reported in 8% (1/13) of women [13]. Moreover, 0.1% (1/725) involved stillbirth, 3.2% (23/724) had a small size for gestational age, and 2.2% (16/724) had congenital anomalies. No cases of neonatal death were reported [12]. Upon statistical analysis, NICU admission (p = 0.5821), Apgar score at 5 min <7 (0.7617), hypoxic ischemic encephalopathy (p = 1), stillbirth (p = 1), and low birth weight (p = 0.5321) or very low birth weight (p = 0.2332) did not differ significantly between vaccinated and unvaccinated women [21]. Other neonatal outcomes were compared between vaccinated and unvaccinated pregnant women: intrauterine growth restriction [0.5% (36/7,530) vs. 0.5% (38/7,530)], stillbirth [1/7,530 (<0.1) vs. 2/7,530 (<0.1)], and preterm birth (<37 weeks) [77/1,387 (5.6%) vs. 85/1,427 (6.0%)]. Additionally, median birth weeks (IQR) were 39 (38–40) vs. 39 (38–40); and median infant weights (IQR) were 3.2 (2.9–3.6) kg vs. 3.2 (2.9–3.5) kg [24].

## Discussion

### Main findings

Two studies demonstrated the efficacy of the Pfizer–BioNTech and Moderna vaccines for preventing future SARS-CoV-2 infection in pregnant women [21, 24]. Following vaccinations with the Pfizer–BioNTech and Moderna vaccines, the vast majority of pregnant women had injection-site pain or soreness [12, 13, 18]. The most common systemic adverse events were fatigue, headache, chills, myalgia, fever, and nausea [12, 18]. There were stark differences in the adverse events reported between the study by Goldshtein et al. [24] and other studies [12, 13, 18]. This was likely due to the different methodologies used. That study from Israel used International Classification of Diseases codes, followed by manual chart review for codes for suspected adverse events. That could have resulted in smaller numbers of complaints than in self-reported methods such as the use of v-safe software. The incidence of these systemic adverse events was higher after the second dose than after the first dose [12–14]. Numerically, more individuals experienced systemic adverse events in the Moderna vaccine group than in the Pfizer–BioNTech group [12]. The rates of adverse events did not differ significantly between pregnant and non-pregnant women, with the exception of seizures. However, the woman affected by seizures was known to have a history of seizure disorder and the anticonvulsant level was measured as borderline low [21]. Interestingly, the maternal and neonatal outcomes did not differ between vaccinated and unvaccinated pregnant women [21, 24].

Maternal antibody responses have been reported to develop following vaccination. Upon vaccination, antibody responses were rapidly established, while through infection they formed gradually [22]. The IgG and IgM titers against SARS-CoV-2 were significantly increased after vaccination. The response was also increased after a booster vaccination was given [13, 15]. Although the vast majority of pregnant women exhibited IgG seroconversion, IgM seroconversion was observed in a minority of pregnant women [20]. After vaccination, IgG against S1,

S2, and RBD formed, while IgG against S1, S2, RBD, and N protein formed following natural infections. Moreover, S1 and RBD IgG levels were found to be higher in vaccinated pregnant women. Meanwhile, S2 and N IgG levels were found to be higher in naturally infected pregnant women [22]. Furthermore, RBD IgG and neutralizing antibody levels were higher in vaccinated individuals than in naturally infected ones [14].

Transplacental antibody transfer was also reported. Cord blood antibody and maternal antibody levels were reported to be approximately equal [20]. Additionally, latency and number of doses were correlated with the intensity of transplacental antibody transfer [16, 20]. The longer the latency, the better the transplacental antibody transfer and the higher the IgG. The offspring of mothers receiving two doses of vaccine were also shown to have higher IgG levels [20]. Finally, there was no difference in maternal-fetal IgG response between infected and vaccinated cases [22].

## Findings in other studies

Randomized controlled trials showed that two-dose regimens of both Moderna and Pfizer–BioNTech vaccines provided excellent efficacy at preventing Covid-19 illness, being 94.1% and 95%, respectively, with no associated safety concerns [25, 26]. As reported in a pregnancy cohort [12], more frequent systemic adverse reactions after the second dose of vaccine were also found in non-pregnant individuals [25, 26]. However, these adverse reactions were only transient, being resolved within a few days [25, 26].

The reported spontaneous abortion rate of 12.6% of completed pregnancies within the vaccinated group was still within an acceptable range, since their underlying medical conditions were unknown [27]. For example, Center for Disease Control and Prevention surveillance in 2015 reported an abortion rate of 188 in 1,000 live births [28]. Some vaccinations, including the tetanus toxoid, reduced diphtheria toxoid, and acellular pertussis (Tdap) and killed influenza vaccines, administered to pregnant women showed no increased risk of adverse outcomes [29, 30]. The Tdap vaccine was most often given in the first trimester of pregnancy [29]. Meanwhile, the majority of participants in Covid-19 vaccination studies were in the third trimester of pregnancy [15, 17, 19–23]. The optimal timing of administering Covid-19 vaccine during pregnancy has remained unclear. Although longer latency was associated with better transplacental antibody transfer [20, 22], data from studies on influenza vaccine showed better immunogenicity and protection upon administration in the third trimester rather than in earlier trimesters [31].

Passive immunity in neonates can potentially protect against SARS-CoV-2 infection. However, this passive immunity may change due to placental sieving [32], depending on the gestational age at first vaccination or infection [20, 22]. In Covid-19 infection, poor transplacental antibody transfer was exclusively observed only in the third trimester of pregnancy, even though the maternal antibody response was significantly higher [33]. Interestingly, lower transplacental antibody transfer upon natural viral infection, such as in Zika and Dengue Viruses [34, 35], compared to vaccinations, such as in pertussis and influenza vaccines [36–38], was also observed in SARS-CoV-2 [20, 39].

High vaccination coverage is important to achieve a sufficient threshold for herd immunity in a population, which provides indirect protection for susceptible individuals from infected hosts [40]. This threshold varies across different infections and populations [40–42]. Additionally, a lower threshold would require higher vaccine efficacy [43, 44]. This issue is important since antibody resistance among SARS-CoV-2 variants of concern was reported [45]. Moreover, mutations in some variants of concern changed the transmissibility, severity, and treatment efficacy of Covid-19, especially for neutralizing monoclonal antibody treatments due to immune escape [46–51].

Evidence of reduced binding affinity of neutralizing antibodies against some variants of concern was reported. For the Pfizer–BioNTech and Moderna vaccines, antibody affinity was reported to be 3.5- and 6-fold lower for B.1.1.7 and B.1.351, respectively [14]. With two vaccine doses, although the effectiveness of Pfizer–BioNTech and Moderna vaccines was reported to be reduced, these vaccines still provide excellent protection against B.1.1.7 and B.1.351 [25, 26, 52–54], and even against P.1 and B.1.617.2 variants [53, 55]. Maximizing the coverage of second vaccination would provide stronger protection against these SARS-CoV-2 variants [52].

## Strengths and limitations

This systematic review used the recent available evidence to describe the efficacy, safety, and immunogenicity of Covid-19 mRNA vaccine in pregnancy. All studies included in this review were assessed as being of high quality. However, they were all observational studies due to the lack of reports about RCT on Covid-19 vaccination for pregnant women. These studies reported only on mRNA-type vaccines. Additionally, some major design differences included the number of patients, sample collection methods, and outcome definitions. Moreover, all available studies that were included were only from the United States and Israel.

## Conclusion

This study suggests that mRNA vaccines, especially Pfizer–BioNTech and Moderna vaccines, can reduce the risk of future SARS-CoV-2 infections. These vaccines can induce antibody responses for pregnant women and their fetuses. Pregnant women should be given two doses of vaccine for more robust maternal and fetal antibody responses. Longer latency was associated with a more robust fetal antibody response. The majority of pregnant women who received the vaccination, as either the first or the second dose, would experience injection-site pain. Furthermore, the second dose of the vaccine would produce more systemic adverse events than the first one, and the administration of the Moderna vaccine is more often associated with systemic adverse events. Biologically speaking, we may conclude that vaccination does not affect pregnancy, delivery, or neonatal outcomes in the short term.

## Supporting information

**S1 Checklist. PRISMA 2020 checklist.**
(DOCX)

**S1 File. Literature search.**
(DOCX)

**S1 Table. Newcastle-Ottawa Scale (NOS) quality assessment of each included cohort study.**
(DOCX)

**S2 Table. Joanna Briggs Institute (JBI) critical appraisal for case series study.**
(DOCX)

**S3 Table. Joanna Briggs Institute (JBI) critical appraisal for case report study.**
(DOCX)

**S4 Table. Joanna Briggs Institute (JBI) critical appraisal for cross-sectional study.**
(DOCX)

## Author Contributions

**Conceptualization:** Nando Reza Pratama, Ifan Ali Wafa, David Setyo Budi, Manesha Putra, Citrawati Dyah Kencono Wungu.

**Data curation:** Nando Reza Pratama, Ifan Ali Wafa, David Setyo Budi.

**Formal analysis:** Nando Reza Pratama, Ifan Ali Wafa, David Setyo Budi.

**Methodology:** Nando Reza Pratama, Ifan Ali Wafa, David Setyo Budi, Manesha Putra, Manggala Pasca Wardhana.

**Supervision:** Citrawati Dyah Kencono Wungu.

**Validation:** Manesha Putra, Manggala Pasca Wardhana, Citrawati Dyah Kencono Wungu.

**Writing – original draft:** Nando Reza Pratama, Ifan Ali Wafa, David Setyo Budi.

**Writing – review & editing:** Manesha Putra, Manggala Pasca Wardhana, Citrawati Dyah Kencono Wungu.

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
