## [Decision Letter · Decision Letter 0]

30 Sep 2021

PONE-D-21-23400mRNA Covid-19 Vaccines in Pregnancy: A Systematic ReviewPLOS ONE

Dear Dr. Wungu,

Thank you for submitting your manuscript to PLOS ONE. After careful consideration, we feel that it has merit but does not fully meet PLOS ONE’s publication criteria as it currently stands. Therefore, we invite you to submit a revised version of the manuscript that addresses the points raised during the review process.

We look forward to receiving your revised manuscript.

Kind regards,

Linglin Xie

Academic Editor

PLOS ONE

Journal Requirements:

2. We note that this manuscript is a systematic review or meta-analysis; our author guidelines therefore require that you use PRISMA guidance to help improve reporting quality of this type of study. Please upload copies of the completed PRISMA checklist as Supporting Information with a file name “PRISMA checklist

Reviewers' comments:

Reviewer's Responses to Questions

**Comments to the Author**

1. Is the manuscript technically sound, and do the data support the conclusions?

Reviewer #1: Yes

2. Has the statistical analysis been performed appropriately and rigorously? 

Reviewer #1: Yes

3. Have the authors made all data underlying the findings in their manuscript fully available?

Reviewer #1: Yes

4. Is the manuscript presented in an intelligible fashion and written in standard English?

Reviewer #1: No

5. Review Comments to the Author

Reviewer #1: GENERAL : There are several grammatical errors and wrong use of tenses that make significant portions of the text difficult to understand.

ABSTRACT: Page 2, under methods, line 38 EU should be written in full 1st before abbreviating. Throughout the text abbreviations should 1st be written in full. In lines 39-40 what do the authors mean by he databases were queried? Lines 61-62 should be recasted. Grammatical errors and wrong use of tenses.

INTRODUCTION: Page 3,line 69, which pandemic is being referred to? The statement in lines 74-75 about the risk of severe COVID 19 in pregnancy seems to be at variance with the 1st sentence in the objective section of the abstract. The term "pregnant women should be substituted for pregnant people throughout the text. The aim of the study in page 3 lines 87-90 should be recasted.

METHODS: Several grammatical errors. Under inclusion criteria, lines 100-101 page 4, number 4 is no clear. Where studies that were not conducted in English included? Page 5,under statistical analysis, pages 139-140 is not clear.

RESULTS: Page 5,lines 144 is not clear, Are 4.018 and 200 both all the records that were yielded? Several grammatical errors. Under outcomes, page 6 lines 168-171 is not clear,

DISCUSSION: TOO many grammatical errors and wrong use of tenses.

6. PLOS authors have the option to publish the peer review history of their article (what does this mean?). If published, this will include your full peer review and any attached files.

Reviewer #1: No

---

## [Author Response · Author response to Decision Letter 0]

22 Oct 2021

Responses to the reviewer #1 comments:

We would like to start by thanking the reviewers for careful and constructive comments and questions. It prompted us to change some of the aspects of our original manuscript, mainly the introduction and discussion along with the conclusion. We strongly believe that our manuscript is much better now than it was before the rewriting.

After careful reading of the comments, we decided that many parts of the article definitely need improvements, especially those with grammatical errors and wrong use of tenses. Also, the description of abbreviations within the table has been attached below each table. We tried to incorporate all suggestions made by the reviewer in this new version of our manuscript. 

We decided to err on the side of caution and fix the language issue of our manuscript this time.

Reviewer #1: GENERAL : There are several grammatical errors and wrong use of tenses that make significant portions of the text difficult to understand.

• We have modified the previous manuscript into the revised manuscript with the more appropriate writing English this time. We also got a proofread certificate from a language editing center (Enago). We hope that this at least partially addresses the reviewer's concerns about some issues in our manuscript. 

ABSTRACT: Page 2, under methods, line 38 EU should be written in full 1st before abbreviating. Throughout the text abbreviations should 1st be written in full. In lines 39-40 what do the authors mean by he databases were queried? Lines 61-62 should be recasted. Grammatical errors and wrong use of tenses.

• We have replaced European Union instead of the EU in the abstract. 

• The word queried here means that we conducted the search in the databases mentioned in the abstract. To make it clearer, we have changed the word ‘queried’ into ‘systematically search’. 

• We have recasted this sentence. If you think it is still not suitable, please any feedbacks are welcomed.

INTRODUCTION: Page 3,line 69, which pandemic is being referred to? The statement in lines 74-75 about the risk of severe COVID 19 in pregnancy seems to be at variance with the 1st sentence in the objective section of the abstract. The term "pregnant women should be substituted for pregnant people throughout the text. The aim of the study in page 3 lines 87-90 should be recasted.

• The term pandemic refers to the Covid-19 pandemic and we have changed ‘Covid-19 pandemic’ in this revision.

• We agree with the reviewer which this sentence seems to be at variance with the first sentence in the objective. Since pregnancy has been known to be a factor that leads to poorer outcomes in Covid-19 infection, we decided to revise this sentence so that it would be linear to our objective in the abstract.

• Recently, some opinions suggest the use of pregnant people instead of pregnant women. For this reason, we used this term in our previous manuscript. However, we consider changing the term pregnant people throughout the texts as per the reviewer's suggestion in this revised manuscript.

• We have recast the aim and integrated it into the end of the second paragraph.

METHODS: Several grammatical errors. Under inclusion criteria, lines 100-101 page 4, number 4 is no clear. Where studies that were not conducted in English included? Page 5,under statistical analysis, pages 139-140 is not clear.

• Some studies do not report the minimum age of the participants, but we decided to include these articles. As a result, we think that it would be more appropriate to just write pregnant women in the inclusion criteria, without restricting the age.

• Criterion no 4 in the previous manuscript has been modified into a clearer sentence. All studies were reported in English. Therefore, we changed ‘no language restriction’ (at line 110 of the previous manuscript) in order to be in line with the inclusion criteria.

• We initially borrow the term Synthesis Without Meta-analysis (SWiM) from PRISMA guideline, for use in systematic reviews examining the quantitative effects of interventions for which meta-analysis of effect estimates is not possible, or not appropriate, for a least some outcomes. Because this review had already adhered to PRISMA guidelines 2020, we decided to omit the SWiM term and simplify this sentence to make it clearer in the revised manuscript.

RESULTS: Page 5,lines 144 is not clear, Are 4.018 and 200 both all the records that were yielded? Several grammatical errors. Under outcomes, page 6 lines 168-171 is not clear,

• In this revised manuscript, the number of studies obtained from our search has been combined into 4218, instead.

• The vaccine efficacy was represented by the infection rate in which the lower infection rate indicates a higher efficacy. We are trying to emphasize that some studies evaluate the efficacy in a different way, i.e. different grouping. In this revised manuscript, we have attempted to improve the grammatical errors.

DISCUSSION: TOO many grammatical errors and wrong use of tenses.

• We have reworked the discussion to justify the writing style with more appropriate use of grammar and tenses as per the reviewer's suggestion. Any feedbacks are welcome if necessary.

---

## [Decision Letter · Decision Letter 1]

23 Nov 2021

PONE-D-21-23400R1mRNA Covid-19 Vaccines in Pregnancy: A Systematic ReviewPLOS ONE

Dear Dr. Wungu,

Thank you for submitting your manuscript to PLOS ONE. After careful consideration, we feel that it has merit but does not fully meet PLOS ONE’s publication criteria as it currently stands. Therefore, we invite you to submit a revised version of the manuscript that addresses the points raised during the review process.

We look forward to receiving your revised manuscript.

Kind regards,

Linglin Xie

Academic Editor

PLOS ONE

Journal Requirements:

Reviewers' comments:

Reviewer's Responses to Questions

**Comments to the Author**

1. If the authors have adequately addressed your comments raised in a previous round of review and you feel that this manuscript is now acceptable for publication, you may indicate that here to bypass the “Comments to the Author” section, enter your conflict of interest statement in the “Confidential to Editor” section, and submit your "Accept" recommendation.

Reviewer #1: (No Response)

2. Is the manuscript technically sound, and do the data support the conclusions?

Reviewer #1: Yes

3. Has the statistical analysis been performed appropriately and rigorously? 

Reviewer #1: I Don't Know

4. Have the authors made all data underlying the findings in their manuscript fully available?

Reviewer #1: Yes

5. Is the manuscript presented in an intelligible fashion and written in standard English?

Reviewer #1: Yes

6. Review Comments to the Author

Reviewer #1: METHODS: Line 83should be --maternal antibody titre instead of maternal titre antibodies. Likewise, line 85 should be cord blood antigen titre. Under search strategy87, it is not clear if all the studies used for this research were conducted between June 20th and 22nd 2020--line 90-91. Lines 142-143 is not clear, was the median age of the women actually estimated in weeks? Lines 178-179 is not clear.

DISCUSSION: The following should be written in full 1st--CDC, ICD, Tdap, DENV, RCTS. Lines 324-326 is not clear, did the authors actually assess pregnancy, delivery and neonatal outcomes in non pregnant women?

7. PLOS authors have the option to publish the peer review history of their article (what does this mean?). If published, this will include your full peer review and any attached files.

Reviewer #1: No

---

## [Author Response · Author response to Decision Letter 1]

25 Nov 2021

Responses to the reviewer #1 comments:

We would like to start by thanking the reviewers for careful and constructive comments and questions. We strongly believe that our manuscript is much better now than it was before the rewriting.

Reviewer #1: 

METHODS: Line 83should be --maternal antibody titre instead of maternal titre antibodies. Likewise, line 85 should be cord blood antigen titre. 

• Thanks for the suggestion. We have incorporated your suggestion into our manuscript.

Under search strategy87, it is not clear if all the studies used for this research were conducted between June 20th and 22nd 2020--line 90-91. 

• We really appreciate this very useful feedback. We actually performed the search between 20th and 22nd of June, 2021 and include all articles before these date. We have fixed the sentence and we hope the meaning will be clearer after this.

Lines 142-143 is not clear, was the median age of the women actually estimated in weeks? 

• Thank you for addressing this issue. We actually mean it as gestational age, not age of the mother. Therefore, we use ‘gestational age’ this time.

Lines 178-179 is not clear.

• Thank you for the insightful feedback. We have paraphrased this sentence, thus it should have a clearer meaning.

DISCUSSION: The following should be written in full 1st--CDC, ICD, Tdap, DENV, RCTS. 

• Thank you for the meticulous observation. We have corrected this issue and all of these abbreviations have been written in full at 1st.

Lines 324-326 is not clear, did the authors actually assess pregnancy, delivery and neonatal outcomes in non pregnant women?

• Thank you for spotting this mistake. We realized that we have mistaken to write the comparison as pregnant vs non-pregnant. Indeed, the reference evaluated the outcomes between vaccinated and unvaccinated pregnant women. Additionally, since we have defined pregnancy and delivery outcomes as maternal outcomes, we simplify the outcomes to be maternal outcomes and neonatal outcomes, instead of pregnancy, delivery and neonatal outcomes.

---

## [Editor Report · Decision Letter 2]

1 Dec 2021

mRNA Covid-19 Vaccines in Pregnancy: A Systematic Review

PONE-D-21-23400R2

Dear Dr. Wungu,

We’re pleased to inform you that your manuscript has been judged scientifically suitable for publication and will be formally accepted for publication once it meets all outstanding technical requirements.

Kind regards,

Linglin Xie

Academic Editor

PLOS ONE
---

## [Editor Report · Acceptance letter]

6 Dec 2021

PONE-D-21-23400R2 

mRNA Covid-19 Vaccines in Pregnancy: A Systematic Review 

Dear Dr. Wungu:

I'm pleased to inform you that your manuscript has been deemed suitable for publication in PLOS ONE. Congratulations! Your manuscript is now with our production department. 

Kind regards, 

on behalf of

Dr. Linglin Xie 

Academic Editor

PLOS ONE